**Data Availability Statement:** All relevant data are within the manuscript and its Supporting Information files.

# Suppressive impact of metronomic chemotherapy using UFT and/or cyclophosphamide on mediators of breast cancer dissemination and invasion

**Raquel Muñoz**[1¤☯]*, **Denise Hileeto**[2☯], **William Cruz-Muñoz**[1☯], **Geoffrey A. Wood**[3], **Ping Xu**[1], **Shan Man**[1], **Alicia Viloria-Petit**[4‡], **Robert S. Kerbel**[1,5‡]*

**1** Biological Sciences Platform, Sunnybrook Research Institute, Toronto, Canada, **2** School of Optometry & Vision Science, University of Waterloo, Waterloo, Ontario, Canada, **3** Department of Pathobiology, Ontario Veterinary College, University of Guelph, Guelph, Ontario, Canada, **4** Department of Biomedical Sciences, Ontario Veterinary College, University of Guelph, Guelph, Ontario, Canada, **5** Department of Medical Biophysics, University of Toronto, Toronto, Canada

☯ These authors contributed equally to this work.
¤ Current address: Department of Biochemistry, Physiology and Molecular Biology, University of Valladolid, Valladolid, Spain.
‡ These authors also contributed equally to this work.
* robert.kerbel@sri.utoronto.ca (RSK); rmunoz@bio.uva.es (RM)

## Abstract

Metronomic chemotherapy using the 5-FU prodrug uracil-tegafur (UFT) and cyclophosphamide (CTX) was previously shown to only modestly delay primary tumor growth, but nevertheless markedly suppressed the development of micro-metastasis in an orthotopic breast cancer xenograft model, using the metastatic variant of the MDA-MB-231 cell line, 231/LM2-4. Furthermore, a remarkable prolongation of survival, with no toxicity, was observed in a model of postsurgical advanced metastatic disease. A question that has remained unanswered is the seemingly selective anti-metastatic mechanisms of action responsible for this treatment. We assessed the *in vivo* effect of metronomic UFT, CTX or their combination, on vascular density, collagen deposition and c-Met (cell mediators or modulators of tumor cell invasion or dissemination) via histochemistry/immunohistochemistry of primary tumor sections. We also assessed the effect of continuous exposure to low and non-toxic doses of active drug metabolites 5-fluorouracil (5-FU), 4-hydroperoxycyclophosphamide (4-HC) or their combination, on 231/LM2-4 cell invasiveness *in vitro*. In the *in vivo* studies, a significant reduction in vascular density and p-Met[Y1003] levels was associated with UFT+CTX treatment. All treatments reduced intratumoral collagen deposition. In the *in vitro* studies, a significant reduction of collagen IV invasion by all treatments was observed. The 3D structures formed by 231/LM2-4 on Matrigel showed a predominantly Mass phenotype under treated conditions and Stellate phenotype in untreated cultures. Taken together, the results suggest the low-dose metronomic chemotherapy regimens tested can suppress several mediators of tumor invasiveness highlighting a new perspective for the anti-metastatic efficacy of metronomic chemotherapy.

**Funding:** This work was supported by grants from Canadian Institutes of Health Research grant 364411 and Canadian Breast Cancer Foundation to RSK, and an Ontario Veterinary College Bull Fellowship to AVP and RM.

**Competing interests:** The authors have declared that no competing interests exist.

## Introduction

An investigational form of therapy known as low-dose metronomic chemotherapy has been studied both preclinically and clinically for almost two decades [1–5]. Metronomic chemotherapy refers to the close regular (continuous) administration of less than maximum tolerated doses (MTDs) with each administration of a conventional chemotherapy drug, generally over long periods, in the absence of any prolonged (e.g. 2–3 week) break periods [1–5]. The proposed main anti-tumor mechanisms mediated by metronomic chemotherapy include inhibition of angiogenesis [1,2,6], stimulation of adaptive T and possibly innate NK cell mediated immunity [7–11] and direct tumor cell killing [12]. The relative impact of these different mechanisms can depend on and vary with the chemotherapy drug used, and likely other variables such as drug dose and schedule [9,10]. A prime example of this is the long-standing observation that low-dose cyclophosphamide (CTX) can induce or augment tumor immunity, especially by targeting immunosuppressive T regulatory (Treg) cells [7].

Most preclinical and metronomic clinical trial studies utilize oral drugs such as CTX, the 5-fluorouracil (5-FU) prodrugs capecitabine or UFT (uracil plus tegafur), methotrexate, topotecan, and vinorelbine [13–20], all of which can be taken on a convenient out-patient basis at home, and are usually associated with reduced serious acute toxic (adverse) events [4]. Furthermore, these drugs, all being off-patent, mean that the cost of metronomic chemotherapies can be particularly inexpensive [5]

A number of preclinical metronomic chemotherapy studies have shown surprisingly potent anti-tumor effects in some cases, even when treating advanced metastatic disease in mice, as well as earlier stage micrometastatic disease [13,14,19]. Such findings have helped contribute to the rationale of undertaking clinical trials assessing metronomic chemotherapy, including some recent randomized phase III clinical trials. Given the presumed lesser toxicity of metronomic chemotherapy and the convenience of oral dosing, most of these phase III trials have evaluated metronomic chemotherapy regimens as long-term maintenance therapies following induction (upfront) conventional MTD chemotherapy regimens. Two noteworthy examples are the Dutch Colorectal Group CAIRO3 trial [18] in first line metastatic colorectal cancer patients, which evaluated daily continuous lower dose capecitabine combined with the VEGF antibody, bevacizumab (Avastin®) as maintenance, compared to observation only (i.e., no further treatment in patients who did not progress after induction therapy), which reported a prolongation of progression free survival [18]. A second phase III clinical trial involved young adult rhabdomyosarcoma patients at high risk for recurrence after induction therapy, utilizing metronomic daily low-dose CTX, plus oral vinorelbine as a follow-up maintenance regimen [20]. This pediatric oncology trial showed a benefit in both progression-free survival and overall survival compared to the observation/no treatment arm, and will become the new standard-of-care for this indication [20]. In addition, a long term (1 year) protocol of maintenance low-dose CTX (50mg daily) with methotrexate twice a week was evaluated as an adjuvant therapy in women with early stage breast cancer, after standard induction therapy, in a randomized phase III trial (ISBCG 00–22); a trend for increased survival was reported with a post hoc analysis indicating a progression free survival and overall survival benefit in the triple negative breast cancer (TNBC) subgroup [21]. Many other phase II breast cancer trials suggest possible benefits of metronomic chemotherapy regimens [22–26], including in TNBC patients [17,27].

In general, more trials of metronomic chemotherapy are being evaluated in breast cancer than any other indication (www.clinicaltrials.gov and above) utilizing a variety of chemotherapy drugs, especially 'doublets' such as vinorelbine with CTX, capecitabine with CTX, and capecitabine with vinorelbine [22–26,17,27]. In this regard, a pooled analysis of six

randomized controlled trials showed that metronomic treatment with UFT, over two years, appears to improve the survival of patients with node-negative breast cancer [28].

Given the encouraging, albeit somewhat limited, recent phase III clinical results of metronomic chemotherapy, and the encouraging multiple phase II clinical trial results in breast cancer over the last 5 years, we decided to resume further preclinical metronomic chemotherapy mechanism-related studies in a TNBC model, the nature of which we first reported over a decade ago. Specifically, using the highly metastatic MDA-MB-231 human breast cancer variant 231/LM2-4 generated by our group [13], we previously evaluated the effects of UFT administered in a daily low-dose oral metronomic schedule, either alone or in combination with metronomic CTX. We examined the effect of these treatments on localized primary orthotopically transplanted tumors, and on the progression of metastases following primary tumor resection. Only treatment with CTX showed delayed tumor growth, while UFT alone was ineffective. However, a marked slowdown in the development of micrometastases was observed in the treated groups. In the experiment involving treatment of advanced metastatic disease, using survival as an end point, UFT therapy alone did not have any significant effect, the CTX monotherapy was, to a small degree, effective in prolonging survival. Nevertheless, the continuous combination therapy of a UFT+CTX doublet for 140 days brought about an almost complete resolution of established (advanced) metastatic disease in the absence of any overt toxicity [13]. In addition, a number of different *in vivo* models that mimic early rather than advanced metastatic disease, have been developed by other groups. In these models, daily, low-dose UFT was effective at controlling micrometastases, not only in breast [29] but also in lung cancer [30,31].

Important information that remains lacking from a majority of the studies showing the effectiveness of low-dose UFT and CTX combination regimens is the anti-tumor mechanistic basis involved. Previous pre-clinical studies indicated anti-angiogenic activity of metronomic treatment with CTX [16,32] and UFT [33,34] when used as individual agents. Several clinical studies have shown a reduction in vascular endothelial growth factor (VEGF) plasma levels associated with the treatment of continuously low-dose CTX alone or in combination [35,36] and by metronomic UFT administration [37,38]. Interestingly, c-Met receptor tyrosine kinase activation was suggested as a mediator of the development of more aggressive tumor phenotypes upon VEGF signaling inhibition [39]. Overexpression and activation of c-Met might occur in response to tumor hypoxia elevation as a compensatory mechanism to VEGF-pathway inhibition, or as a result of the systemic host-mediated response to VEGF inhibitory treatment.

Given this background, we decided to employ our previously established orthotopic 231/LM2-4 breast cancer model to examine the effect of metronomic UFT, CTX and the UFT +CTX combination on vascular density, collagen deposition and c-Met in the primary tumor setting, which we assessed by means of histochemistry/immunohistochemistry. The rationale for restricting these studies to primary tumor treatment models is the observation that despite the modest or even lack of effects on primary tumor growth in terms of changes in tumor volume, there was a marked suppressive effect on development of micrometastases, even by metronomic UFT treatment alone, as noted above. We reasoned that this unexpected effect on micrometastases might be relevant to explaining the potent suppressive effect of metronomic UFT+CTX on established late stage metastatic disease. To complement the *in vivo* findings, and to assess evidence for the possible direct anti-invasive effect of the aforementioned treatments, we also evaluated the effect of continuous low-dose of 5-FU, 4-HC and the 5-FU+4-HC combination on the migratory and invasive potential of the 231/LM2-4 metastatic breast cancer cell line *in vitro*.

Our results indicate that metronomic therapy *in vivo* using the CTX and UFT prodrugs alone, but especially in combination, inhibit several and diverse mediators of tumor dissemination *in vivo* and moreover, their corresponding active metabolites, 5-FU and 4-HC, similarly inhibit tumor cell invasiveness *in vitro*.

## Results

### UFT, CTX and combination UFT+CTX therapy have only limited effect on the growth of 231/LM2-4 primary orthotopic tumors

231/LM2-4 human breast cancer variant cell line was orthotopically implanted in the mammary fat pad of female severe combined immunodeficient (SCID) mice. Treatment with UFT, CTX and the UFT+CTX doublet combination were administered in a metronomic dose/schedule daily, at the previously determined optimal biological doses, and started at day 14 after cell injection, when tumors reached a size of 160 to 190mm$^3$. The effect of the different treatments on primary tumor growth is shown in Fig 1. UFT had no discernible antitumor effect when assessed by changes in tumor size up to day 34 post-cell implantation (One-way ANOVA, Dunnett's comparison P>0.05). At this point vehicle and UFT groups required termination as they had reached end-point (approximate tumor volume of 1,300 mm$^3$). By contrast, CTX and the UFT+CTX combination delayed primary tumor growth such that end-point was not reached until day 47 post-cell implantation. These results are in agreement with our previously published results [13] and also with the apparent minimal or modest antitumor effect of metronomic UFT monotherapy reported by others, both pre-clinically and in clinical trials [31,40]. The antitumor effect of low-dose oral CTX-monotherapy regimens have also been shown in different human tumor xenograft models and in various clinical settings [13,41]

No weight loss or other signs of toxicity were observed in any of the treatment groups (S1 Fig). In order to standardize groups by tumor size, mice treated with 0.1% hydroxypropylmethyl

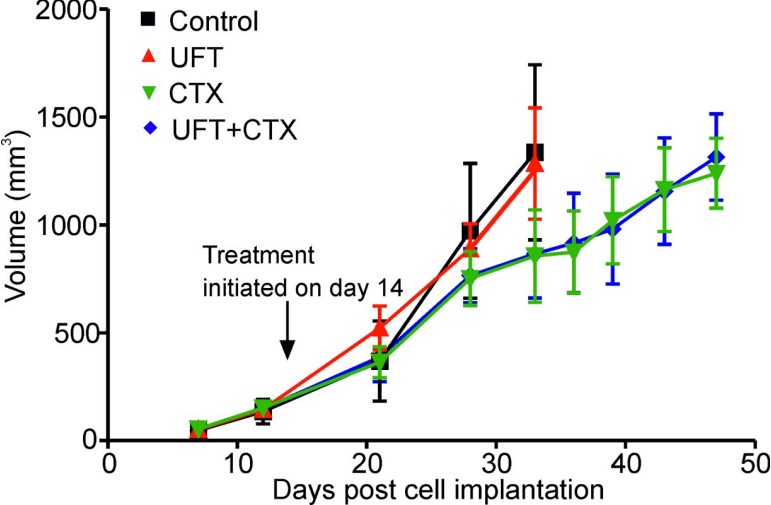

**Fig 1. Effect of UFT, CTX and UFT+CTX metronomic chemotherapy on localized primary tumors.** 231/LM2-4 human breast metastatic variant cells were orthotopically injected into the right inguinal mammary fat pad of 6–8 week-old female SCID mice. When tumors reached volumes of ~160–190 mm$^3$, treatment with vehicle control 0.1% HPMC orally daily (black), 15 mg/Kg/d UFT by gavage (red), 20 mg/Kg/d CTX through the drinking water (green) or the combination UFT+CTX treatment (blue) was initiated. Primary tumors were surgically removed when they reached an average size of approximately 1,300 mm$^3$.

cellulose (HPMC, also referred to as vehicle control) and 15mg/kg/d UFT were sacrificed when their tumor size reached 1,300mm$^3$ and 1,400mm$^3$, respectively, almost three weeks from the start of the treatment. Mice treated with CTX and the UFT+CTX combination were sacrificed when the tumor size reached 1,200mm$^3$ and 1,300mm$^3$ respectively, almost five weeks from commencement of treatment.

## Several biomarkers relevant to tumor cell invasiveness are altered by metronomic chemotherapy using UFT and CTX

After sacrifice, the tumors were removed and prepared for histochemical/immunohistochemical analysis. On the basis of tumor heterogeneity [42], the external border of the tumor (designated as the "invasive border") and intratumoral area, more internal from the external tumor borders (designated as "central region") were analyzed separately.

**Necrosis.** An analysis of hematoxilin and eosin (H&E) stained sections, derived from the central region of both untreated and treated tumors showed large areas of necrosis. The median percentage of necrosis in tumor sections was 78% (70–80% range) for the control group, 80% (70–90% range) for the UFT-treated group, 85% (80–90% range) for the CTX-treated group and 90% in all sections for the group treated with UFT+CTX doublet, as shown in Fig 2A-white columns. A necrotic core region is commonly found in many solid tumors and might be due to the hypoxia and metabolic stress generated by rapid tumor cell growth, vascular collapse and insufficient vascularization in the area [43]. The average percentage of necrosis area was slightly, but not significantly, increased by the treatments. In contrast, major differences between treatment groups were found when analyzing the level of necrosis in the tumor sections at the invasive border. Thus, the median percentage of necrosis was 16% (0–40% range) for untreated tumors, 29% (5–60% range) for the UFT-treated group, 40% (20–80% range) for the CTX-treated group and 66% (20–90% range) for the combination group, as seen in Fig 2A-grey columns. An increase in the percentage median of necrosis of 13%, 24% and 50% was attained by treatments with UFT, CTX and the UFT+CTX combination, respectively. Tumors from animals belonging to the same treatment group showed higher variability in the percentage of necrosis in their invasive border in comparison with their central region.

**Alteration of vascular density.** In order to evaluate the effect of the different treatments on tumor vasculature a double CD31/VEGFR2 staining was performed and the vascular density was analyzed. The tumor blood vessels were distributed in focal clusters (S2 Fig) and revealed a higher concentration at the periphery of tumor sections. The degree of expression of vascular endothelial growth factor receptor 2 (VEGFR2) in tumor vessels has been shown to be variable and dependent on the type of tumor and this fact might have an effect on tumor responses to VEGF pathway inhibiting antiangiogenic drugs [44]. In our 231/LM2-4 tumor xenograft, VEGFR2 was expressed in all vessels regardless of the treatment. All tumor blood vessels were positive for CD31 and for VEGFR2. The average vessel count from the different tumors groups is presented in Fig 2B. Treatment with CTX and with the UFT+CTX combination induced a significant decrease in the density of double positive CD31/VEGFR2 vessels. The decrease in vascular density was observed in both, tumor sections from the central region and from the invasive border. A significant reduction in microvascular density by low dose metronomic CTX treatment in 231/LM2-4 bearing nude mice has been observed previously [45]. However, UFT alone did not cause any reduction in the density of CD31/VEGFR-2-positive tumor blood vessels or tumor volume; instead, on the invasive border of tumors a slight, non-significant increase in vessels was observed (Fig 2B-grey column). This effect was reversed by the combination with CTX. In fact, the decrease in vascular density in the group treated with the UFT+CTX combination was slightly higher than that obtained with CTX monotherapy.

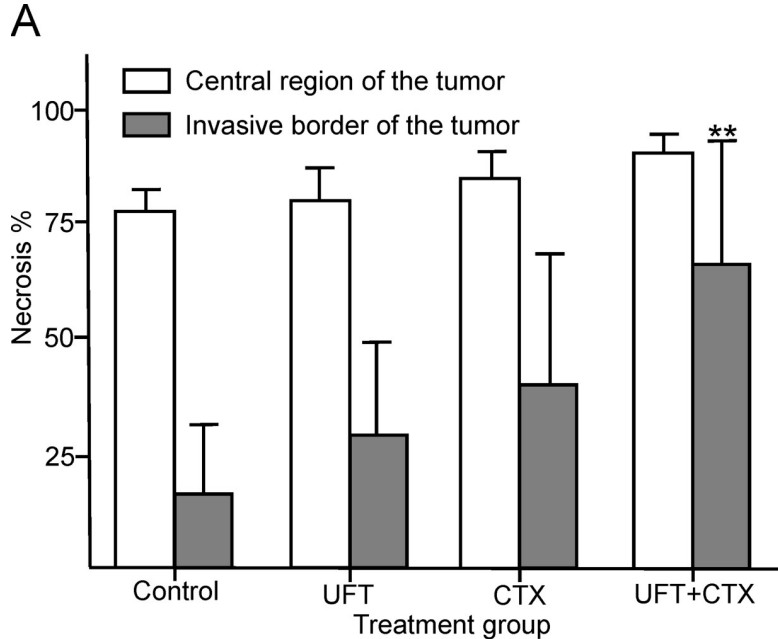

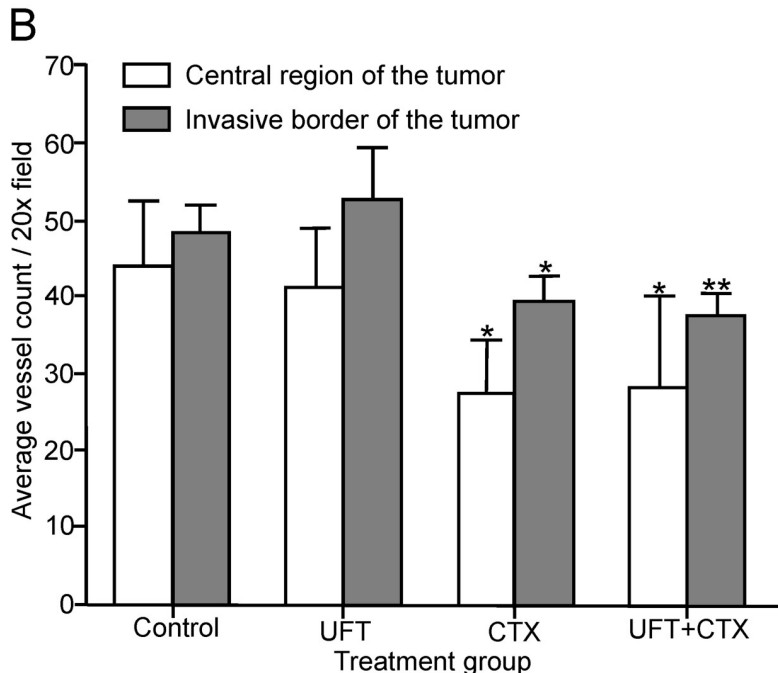

**Fig 2. Tumor necrosis and vascular density are altered by the metronomic chemotherapy treatments.** (A) The percentage of tumor necrosis was assessed by H&E staining in sections of the central region and the invasive border of the untreated and treated tumors. A significant increase in tumor necrosis was observed in the invasive border sections of the UFT+CTX treated tumors (**$p<0.01$ vs. control); significance was analyzed by one-way ANOVA with Dunnett's post-test. (B) Effect of the different treatments on vascular density as determined by double CD31/VEGR2 immunofluorescence labelling. The average vessel count of seven fields per tumor (x20 magnification), from the central region and the invasive border is presented. A significant reduction in vascular density was observed in the CTX (*$p<0.05$ vs. control) and the UFT+CTX (**$p<0.01$ vs. control) groups; significance was analyzed by one-way ANOVA with Dunnett's post-test.

**Local invasion.** The extent of local invasion to soft tissue and deep muscle, assessed by vimentin immunostaining, in the animals corresponding to untreated and treated groups can be seen in Fig 3. Vimentin is a suitable marker to identify the human breast cancer cell line

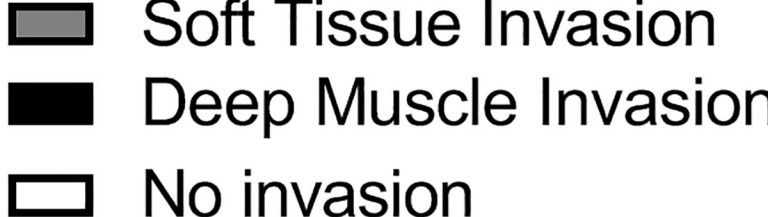

**Fig 3. Analysis of local invasion by vimentin immunostaining in tumor xenograft from all mice.** The percent of mice that showed 231/LM2-4 cells invasion in the adjacent soft tissue (grey), in deep muscle (black) and no invasion (white), at the time of sacrificed, was analyzed by vimentin immunostaining of the tumors from the different treatment groups. The number of animals per group is indicated. The statistical analysis has shown no statistically significant difference in local invasion between groups. A logistic regression was applied for comparing local invasion to either soft tissue or deep muscle for the different treatment groups, and the estimated proportions were calculated using a generalized linear mixed model (GLIMMIX procedure).

231/LM2-4 (S3 Fig) derived from the parental MDA-MB-231, which express basal-like markers and has been classified as a post-epithelial mesenchymal transition cell line, due to the expression of vimentin and the absence of E-cadherin expression [46]. Invasion in adjacent soft tissue was observed in all treatment groups, with lower levels in the drug treated groups. However, no statistically significant difference for the presence of local invasion to soft tissue and deep muscle was observed between treated and control or between treatment groups with P-values (P>0.05) in all cases. The exact P-values were calculated via a logistic procedure, using SAS 9.4 (Cary, NC).

As previously indicated, the tumors from the groups treated with CTX and UFT+CTX were removed two weeks later than the tumors of the untreated and UFT treated animals, i.e., almost five weeks from commencement of the treatment. This standardized the tumors by size to minimize its impact on size-influenced variables, such as tumor hypoxia. At the time of sacrifice, no striking differences were observed among the different groups in relation to deep muscle invasion (Fig 3). Tumor infiltration in the peritoneum was not observed macroscopically in any of the animals.

**Alteration of tumoral collagen deposition.** Collagen is the main structural protein component of the extracellular matrix (ECM), and an increase in collagen deposition and thickness, is often found in regions of tumor cell invasion, indicating that these collagen characteristics might facilitate cancer cell migration and invasion [47]. Bearing in mind these findings, we assessed the presence of intratumoral and peritumoral collagen deposition using Trichrome histochemical staining, in control and treated groups (Fig 4). We used a grading system composed of mild/focal (grade 1), moderated (grade 2), and extensive (grade 3) collagen deposition (S1 Table). The S4 Fig illustrates the difference between the grades. Our results show that there is a trend for lower levels of intratumoral collagen in all treatment groups and for lower peritumoral levels of collagen particularly in UFT+CTX treatment group (Fig 4). These levels, however, do not reach statistical significance (One-Way ANOVA, Dunnett's multiple comparison test P>0.05).

**Reduction of phosphorylated and activated form of c-Met, p-Met[Y1003], by the metronomic chemotherapy treatments.** The 231/LM2-4 cell line was selected, as indicated, from the well characterized MDA-MB-231 human breast cancer cell line, which apart from the typical expression profile of triple negative tumors [48,49], expresses c-Met, cadherin-11 (similar to N-cadherin) and a mutant form of p-53 [50,51].

We examined the expression of c-Met and p-Met[Y1003] by immunohistochemistry in control and treated 231/LM2-4 tumor xenografts. The expression and cellular distribution of c-Met was analyzed using an antibody directed at the cytoplasmic domain of the human c-Met protein. We based this choice on several studies demonstrating that, unlike antibodies directed to the extracellular or transmembrane domain, the use of antibodies directed to the intracellular c-Met domain correlated with poor patient outcome and their use permitted the identification of nuclear c-Met in more aggressive forms of cancer [52,53]. To analyse phosphorylated c-Met, a phosphospecific antibody targeting Y1003 phospho-epitope (localized to the juxtamembrane portion of the receptor) was used. The intensity of c-Met and p-Met[Y1003] staining was evaluated using the following grading system: weak (grade 1), moderate (grade 2) and strong (grade 3) (S2 Table). The tumor necrotic areas were excluded from scoring.

In untreated 231/LM2-4 tumor xenografts, c-Met staining was relatively homogenous throughout the tumor tissue, and was both cytoplasmic and nuclear (Fig 5). The intensity of c-Met cytoplasmic staining was weak (grade 1) increasing to moderate (grade 2) in the tumor periphery. Hyperplastic acini showed cytoplasmic and nuclear staining, displaying strong (grade 3) staining in the nucleus.

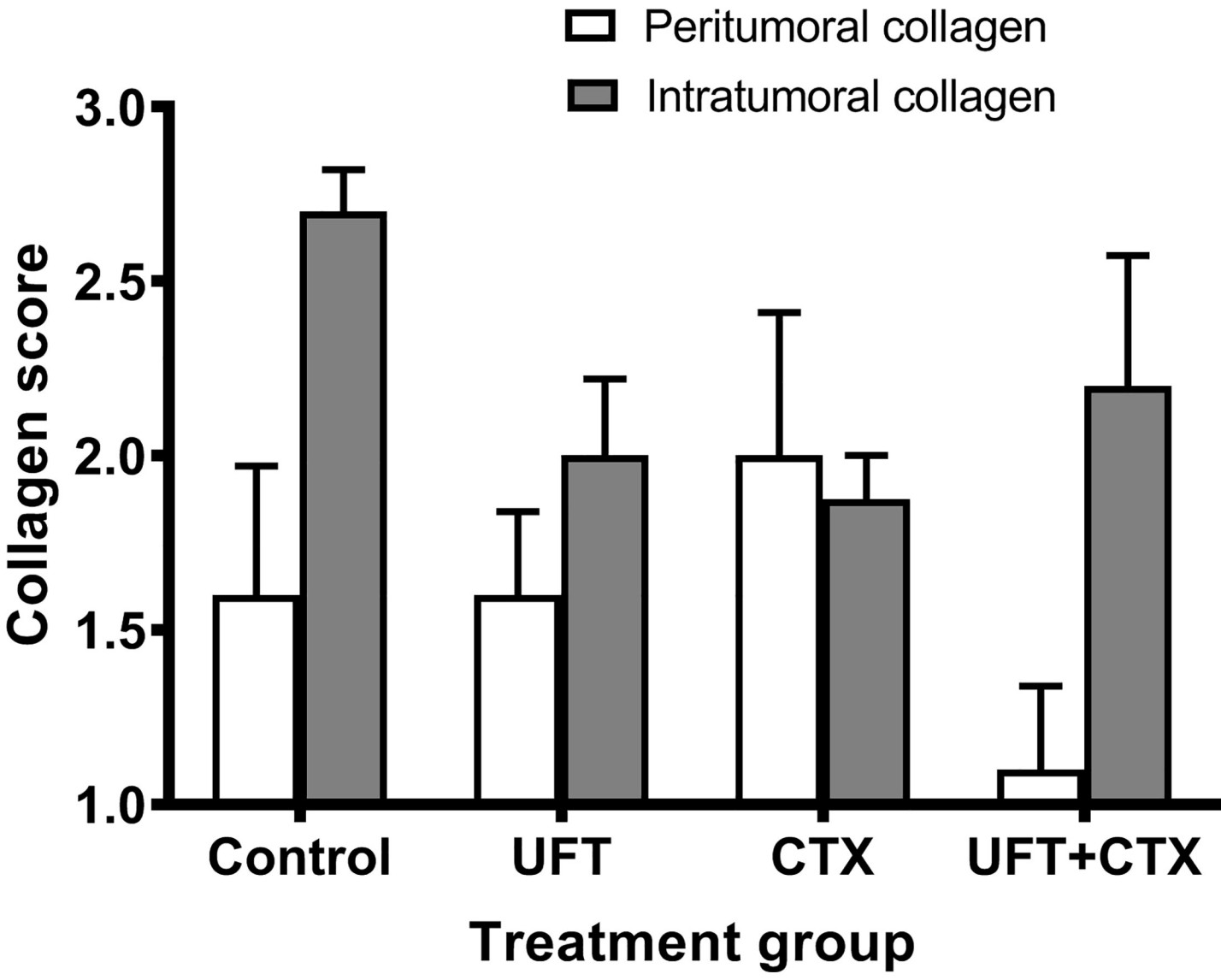

**Fig 4. Assessment of peritumoral and intratumoral collagen deposition in paraffin tumor sections.** The collagen content of the tumors was analyzed by Masson´s trichrome histochemical staining in control and treated groups. A grading system composed of mild/focal (grade 1), moderated (grade 2) and extensive (grade 3) collagen deposition was used. The columns represent the mean ± standard error of the mean (SEM), of the score values corresponding to peritumoral collagen (white) and intratumoral collagen (grey) of the tumor sections corresponding to each treatment group. The statistically analysis by One-Way ANOVA shows no statistical significant difference between groups. (One-Way ANOVA, Dunnett's multiple comparison test $P > 0.05$).

Immunohistochemistry for p-Met[Y1003] in untreated 231/LM2-4 tumor xenografts (Fig 6) demonstrates intense widespread staining in the vital areas with both a cytoplasmic and nuclear pattern. This intensity of cytoplasmic and nuclear staining was moderate (grade 2) increasing to strong (grade 3) at the periphery. In certain tumor areas, a large amount of reactive stroma was present, which stained strong (grade 3) for cytoplasmic and nuclear p-Met [Y1003]; acini surrounded by an invasive tumor showed both cytoplasmic and strong nuclear staining. This suggests that c-Met signalling in "normal" adjacent epithelium and stroma is activated in these aggressive tumors and thus might potentially contribute to tumor progression. Also a significant p-Met[Y1003] nuclear staining was detected in tumor areas of viable cells encircled by necrosis. Of note, overlap of positive p-Met[Y1003] and Ki67 staining was

## c-Met

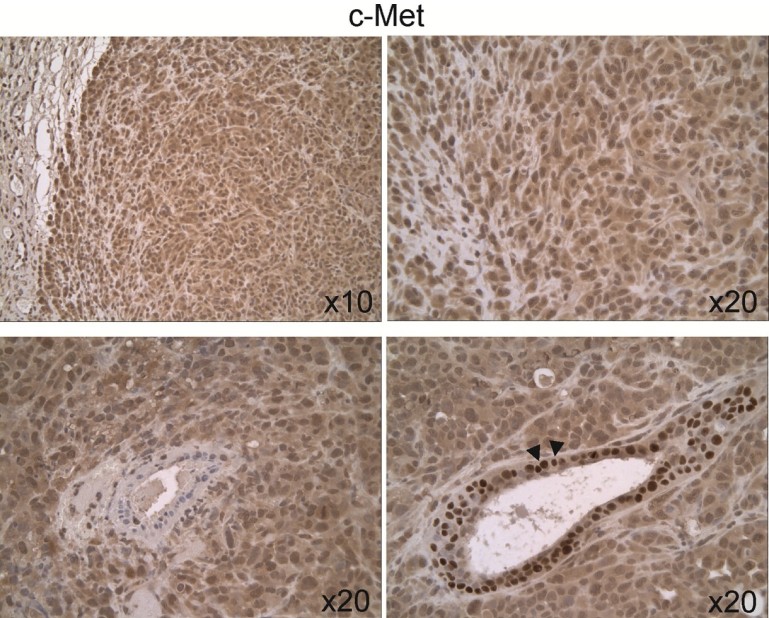

**Fig 5. Expression and cellular distribution of c-Met in untreated 231/LM2-4 tumor xenografts.** c-Met staining was relatively homogenous throughout the tumor tissue, and was both cytoplasmic and nuclear. Hyperplastic acini showed cytoplasmic and nuclear staining, with strong staining in the nucleus (down-right arrows).

found in 231/LM2-4 tumor tissue (S5 Fig). This is supported by a previous study showing nuclear immunoreactivity for the cytoplasmic domain of c-Met only in cells that were not confluent and in active proliferation [54].

## p-Met [Y1003]

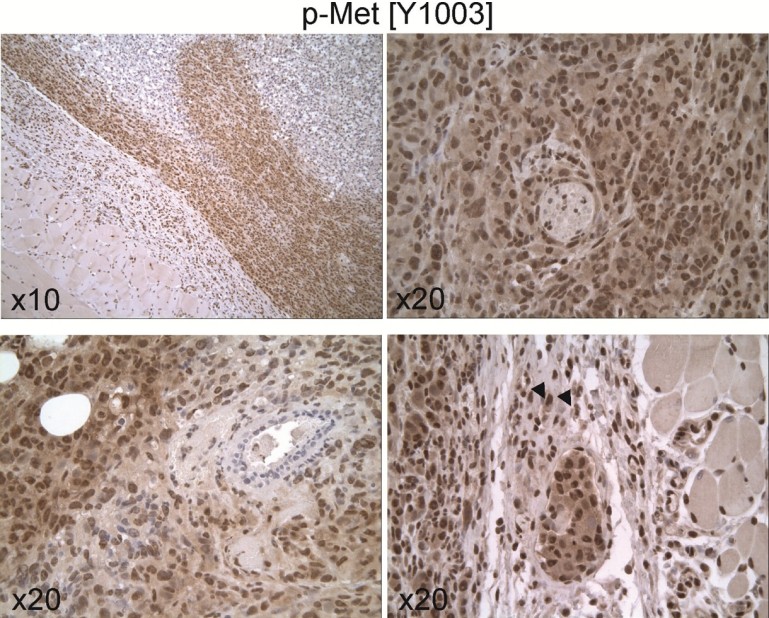

**Fig 6. Expression and cellular distribution of p-Met[Y1003] in untreated 231/LM2-4 tumor xenografts.** Immunohistochemistry for p-Met[Y1003] shows intense widespread staining in the vital areas of the tumor, with both nuclear and cytoplasmic localization. In certain tumor areas, a large amount of reactive stroma was present, which showed strong cytoplasmic and nuclear staining (down-right arrows).

In treated 231/LM2-4 tumor xenografts, strong c-Met levels localizing to the nucleus and cytoplasm were observed in viable tumor areas in all treatment groups, with no noticeable difference (S6 Fig). The staining of p-Met[Y1003] showed only focally immunopositivity in a predominantly nuclear pattern. The average staining scores between untreated and treated groups was as follow: control (2.2)>UFT (2)>CTX (1.75)>UFT+CTX (1.2). The UFT+CTX treatment group showed decreased staining intensity (Fig 7E and 7F) when compared with the control group (Fig 7C and 7D). Using the H scoring system, as specified in Materials and methods, there was a statistically significant decrease in the expression of p-Met[Y1003] in the UFT+CTX treatment group (H-score, 135±6), compared to untreated controls (H-score, 225 ±21). The remaining treatment groups also showed a tendency towards decreased expression of p-Met[Y1003]; however, the differences did not reach statistical significance (Fig 8).

**Continuous exposure to low doses of 5-FU, 4-HC and the 5-FU+4-HC combination variably suppress *in vitro* invasiveness of 231/LM2-4 cells.** In order to study the effect of the active metabolites 5-FU, 4-HC and the 5-FU+4-HC combination treatment on invasive potential of 231/LM2-4 breast cancer cells, we used two *in vitro* three-dimensional (3D) assays as surrogates for *in vivo* tumor cell invasion: a transwell chemoinvasion assay using collagen type IV, and a 3D laminin-rich extracellular matrix (lrECM) "on-top" assay employing Matrigel as a barrier. The concentrations of the drugs used in both assays were 1µM for 5-FU, 0.01µM for 4-HC and 1µM 5-FU+0.01µM 4-HC for drug combination. Continuous treatment of the cells over six days with the aforementioned drugs concentrations had little effect on cell viability (Fig 9A). The viability assay was performed in a two-dimensional (2D) monolayer culture. Previous studies have demonstrated a lower efficacy of chemotherapeutic drugs on several breast tumor cells lines, including MDA-MB-231, cultured in 3D conditions as compared to monolayers [55].

**Transwell invasion assay.** In the transwell chemoinvasion assay using collagen type IV as coating, a significant (p<0.0001) inhibition of migratory and invasion potential of 231/LM2-4 cells was observed in the presence of the drugs *versus* control conditions (Fig 9B).

**Three dimensional *in vitro* assay using Matrigel.** The 3D lrECM "on-top" assay using Matrigel as a barrier seeks to mimic the interstitial matrix *in vivo*, providing an environment where signalling between neighbouring cells and the cell-extracellular matrix is possible [56]. Kenny *et al*. [57] demonstrated that in contrast to the similar morphology adopted by cells growing as 2D monolayer, the culture of epithelial breast cancer cell lines using this 3D assay showed different morphologies, that were classified by analysis with phase contrast and fluorescence microscopy into four groups: Round, Mass, Grape-like and Stellate. The breast cancer cell line MDA-MB-231 was characterized as Stellate showing an invasive phenotype with stellate invasive protrusions. A correlation between 3D cell morphologies and cellular gene and protein expression patterns was observed [57].

In our 3D lrECM "on-top" assay, 231/LM2-4 cells formed multicellular structures. The effect of the different treatments on the morphology of these cellular structures was analyzed by means of phase contrast microscopy. For this purpose, we classified the 3D structures into three different groups, following the previous classification given by Kenny *et al*. [57] with modifications. These groups were Mass, Pseudo-Stellate Mass, and Stellate. Mass structures following a multicellular collective protrusive migration pattern, and containing leading cells with invadopodia structures were denoted as Pseudo-Stellate Mass. None of the structures formed by 231/LM2-4 cells had the previously described Grape-like morphologic phenotype, which is characteristic of cell lines overexpressing the HER-2 protein [57]. Representative examples of the three different morphological phenotypes formed by control cells (Fig 10-left) and treated cells with 1µM 5-FU, 0.01µM 4-HC (S7 Fig) or the combination 1µM 5-FU +

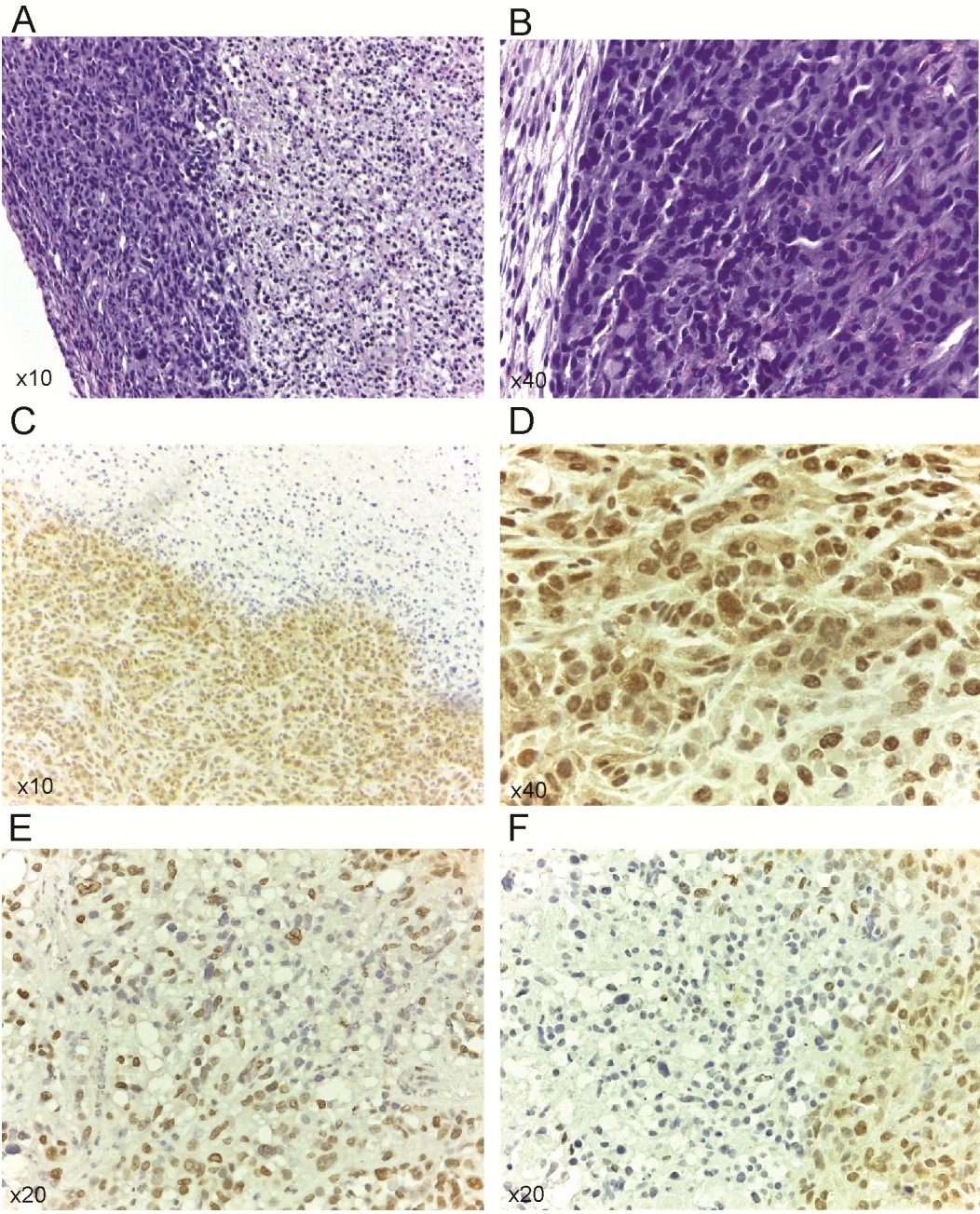

**Fig 7. Expression of p-Met[Y1003] in control and in treated groups.** (A) Tumor section showing peripheral areas with preserved morphology and vast areas of central necrosis. H&E stain. (B) The tumor cells show high pleomorphism, hyperchromasia and increased mitotic activity. H&E stain (C) Immunohistochemistry for p-Met[Y1003]. Untreated controls demonstrate uniform intense immunopositivity in the peripheral vital areas of the tumor (lower left). The necrotic central parts of the tumor are immunonegative (upper right). (D) Untreated controls reveal intense immunopositivity in both nuclear and cytoplasmic distribution. (E, F) UFT+CTX treatment group shows significantly decreased staining intensity with only focal immunopositivity p-Met[Y1003] in a predominantly nuclear pattern. Vast areas of viable tumour are not staining (immunonegative).

0.01µM 4-HC (Fig 10-right) are shown. In Table 1 the percentages of the three morphological phenotypes formed under the different treatments are presented.

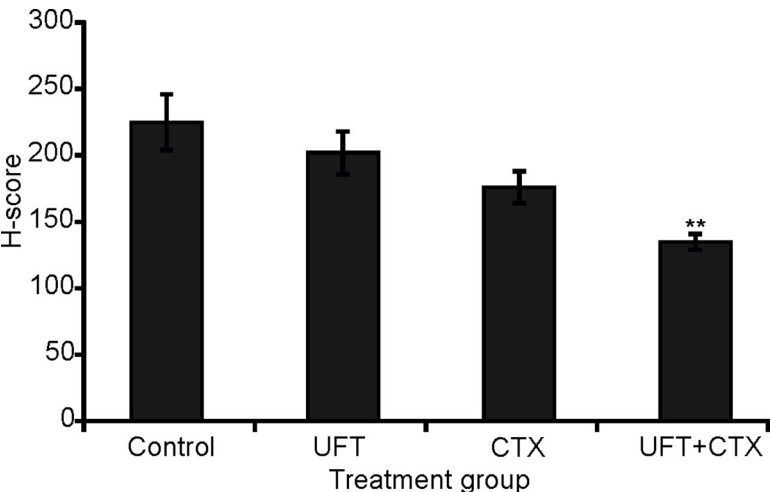

**Fig 8. Immunohistochemistry for p-Met[Y1003] H-score results.** There was a statistically significant decrease in the expression of p-Met[Y1003] in the UFT+CTX treated group, compared to untreated control (Student´s t test, 0.009576 two-tailed p-value).

As this is the first study to assess 3D structures formed by the 231/LM2-4 cells, we conducted a complete morphologic analysis of these structures. As indicated in Table 1, most of the multicellular structures in the control untreated cells display a Stellate phenotype. The structural characteristics of this phenotype are shown in Fig 10C-left and correspond with previous reports [58], it should be noted the presence of several multicellular chains, with a collective cell migration pattern of invasion together with a protrusive leading front with one or multiple leading cells with invadopodia, in which the rear cells follow the leader cell (coordinated migration). The structural characteristic of Mass and Pseudo-Stellate Mass structures formed by control untreated cells are shown in Fig 10A-left and Fig 10B-left respectively. Dissemination of single tumor cells from the structures and collective dissemination were rarely observed.

The Mass and Pseudo-Stellate Mass phenotypes were more abundant following prolonged continuous exposures (6 days) to 5-FU (S7 Fig), 4-HC (S7 Fig) and the 5-FU+4-HC combination (Fig 10-right) than the Stellate phenotype as indicated in Table 1. The structural characteristic of Mass and Pseudo-Stellate Mass structures formed under exposure to 5-FU+4-HC are shown in Fig 10A-right and Fig 10B-right respectively. Mass structures presenting multicellular streaming with no apparent junction contacts were more abundant than in the untreated cells. In all treated cells, an increase in single and collective cell dissemination of tumor cells from Mass and Pseudo-Stellate Mass structures into the Matrigel was observed compared to control.

Interestingly, a different degree of disruption by granularity of some Masses and Pseudo-Stellate Masses was seen in all treated cells. As a consequence, the 3D cluster configuration was either still evident or disappeared, becoming a collection of dispersed single cells. The frequency of these disrupted structures observed by microscope field (x10 magnification) was 0 for untreated cells, 0.5 for 5-FU, 0.7 for 4-HC and 0.55 for 5-FU+4-HC treated cells.

For the fewer Stellate multicellular structures observed in all treated groups, important differences with the control cells were a reduction in the number of multicellular strands per structure, with high variability in their length, and heterogeneity in the leading edge. With regard to the latter, a protrusive leading front was seen, with one or multiple leading cells with invadopodia. As well as, a large number of leading buds, characterized by the absence of a

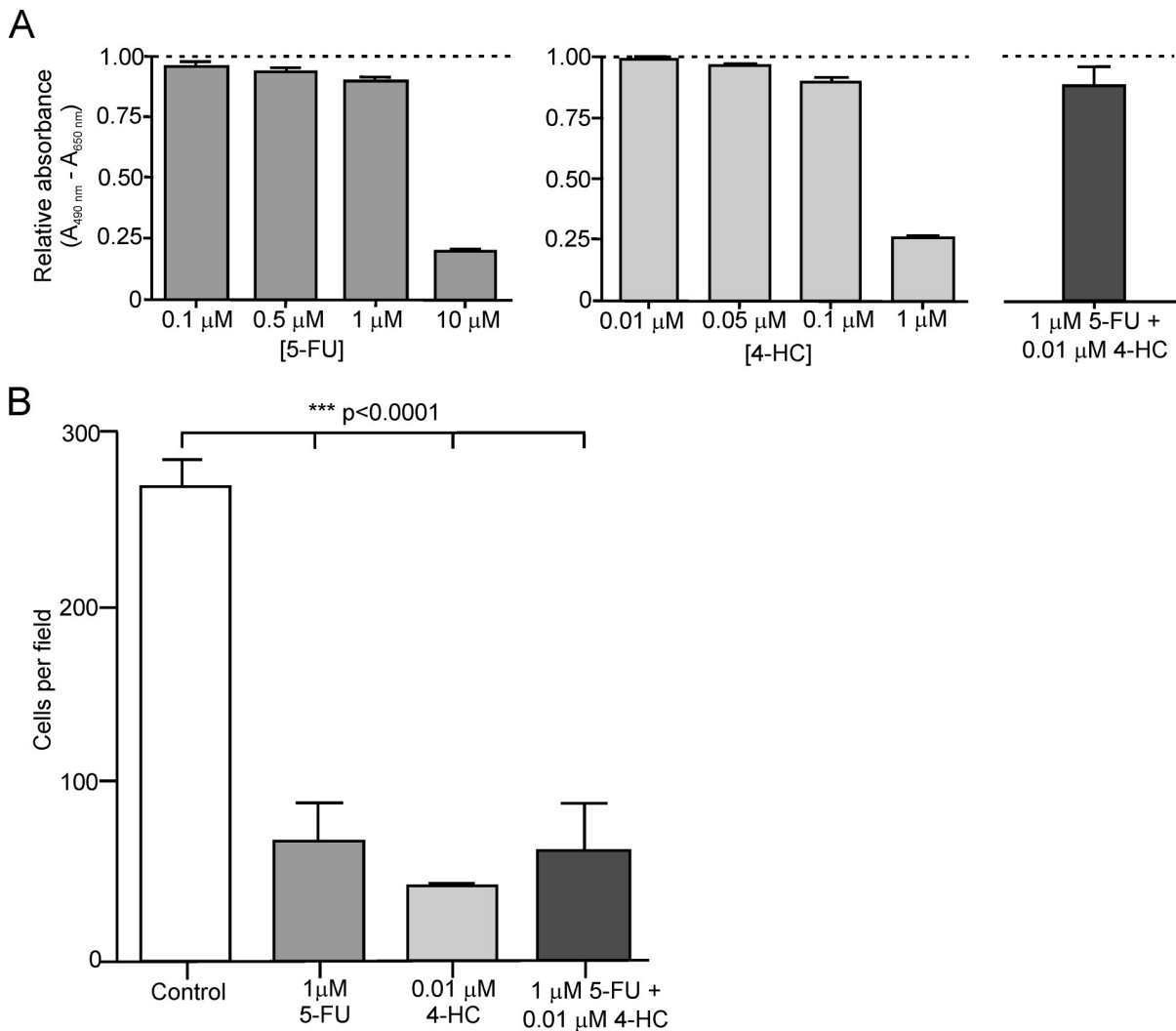

**Fig 9. Effect of continuous exposure to 5-FU, 4-HC and 5-FU+4-HC on viability and invasiveness of 231/LM2-4 cells.** (A) Effect of prolonged continuous exposures (six days) to different concentrations of 5-FU (0.1, 0.5, 1 and 10μM), 4-HC (0.01, 0.05, 0.1 and 1μM), and to the combination 1μM 5-FU + 0.01μM 4-HC, on *in vitro* viability of 231/LM2-4 cells. The relative absorbance ($A_{490nm}$-$A_{650nm}$) to control is represented; $A_{490nm}$ is directly proportional to the number of living cells and $A_{650nm}$ eliminates any kind of background. (B) Assessment of the invasive capacity of 231/LM2-4 cells, under untreated and treated conditions, in a transwell chemoinvasion assay using collagen type IV as coating. The protracted treatments with low dose and non-toxic doses of 1μM 5-FU, 0.01μM 4-HC and the 1μM 5-FU + 0.01μM 4-HC combination are detailed in Materials and methods. Cells that invaded and passed through membrane pores were stained with 0.1% crystal violet and counted. The numbers are the mean ± standard error of the mean (SEM), of the cell counts of eight random fields (x20 magnification) per sample of a representative experiment. The experiment was repeated three times. A significant reduction in invasive capacity of the cells was observed in all treated groups (\*\*\*p<0.0001 *vs.* control). Significance was analyzed by one-way ANOVA with Dunnett's post-test.

leader cell with a well-defined position, and by a smooth rim lacking invadopodia. Thus in the 5-FU+4-HC treated cells (Fig 10C-right), an obvious reduction in the number of strands by structure was observed compared to the control cells. In addition, an uncoordinated arrangement and directionality of the component cells was observed in some multicellular chains, as well as contacts or more commonly fusions between different structures to form a large stellate structure.

A Stellate morphology of 3D structures formed by mesenchymal-like breast cancer cells on Matrigel, including MDA-MB-231, has been associated with invasiveness and metastatic

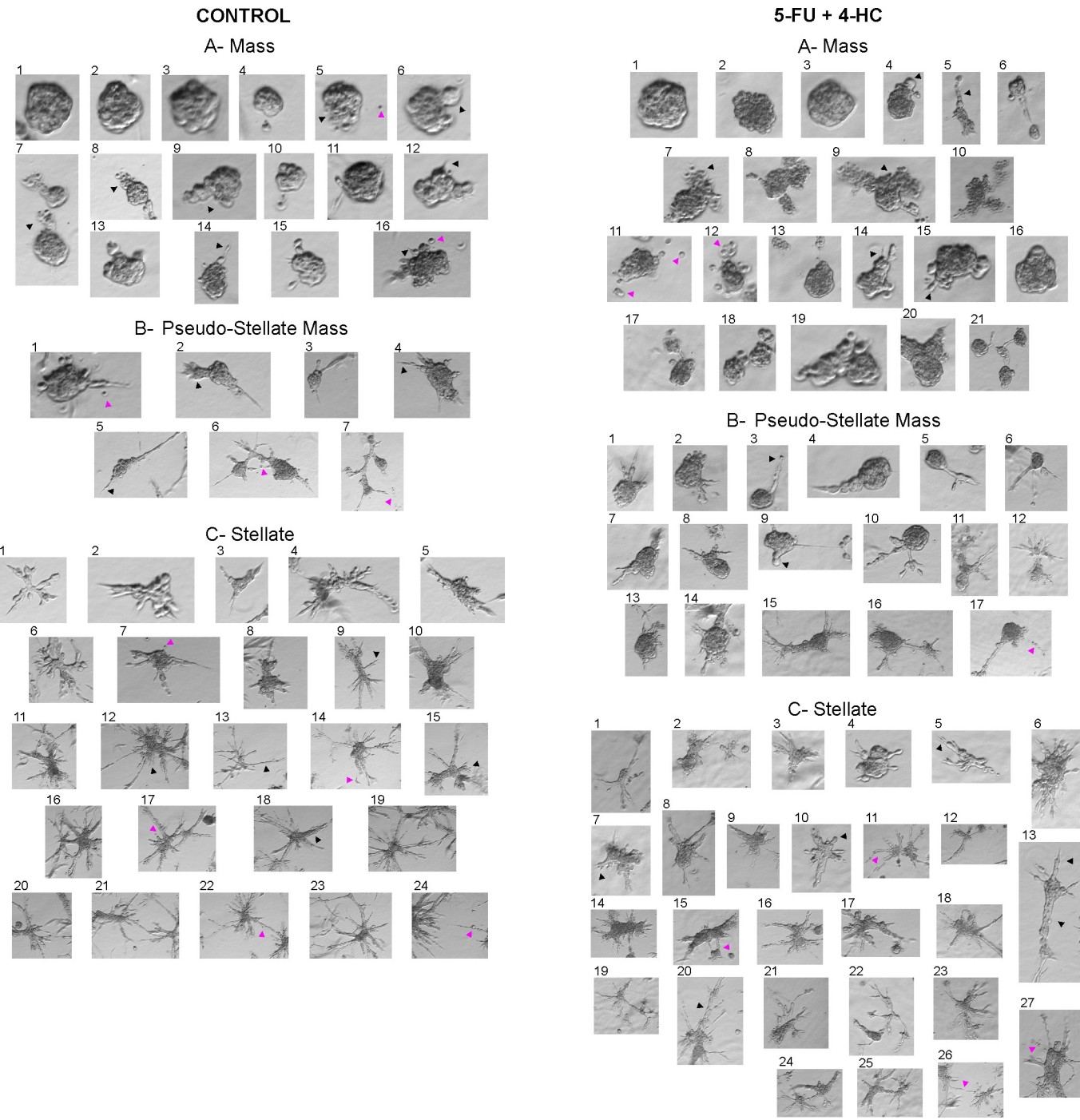

**Fig 10. Invasive capacity of 231/LM2-4 cells under control and 5-FU+4-HC conditions using a 3D lrECM "on-top" assay.** (**Control**): Mass structures (A): round morphology (1–3), irregular morphology showing collective cell migration as chains of few cells with smooth borders (Δ9), buds (Δ6) or as disorganized masses (Δ5). Single cell protrusions (Δ12,14,16). Multicellular streaming with no apparent junction contacts (Δ7,8). Dissemination of single tumor cells (pink Δ5,16). Pseudo-Stellate Mass structures (B): multicellular collective protrusive migration with leading cells with invadopodia (Δ2,4,5). Dissemination of single tumor cells (pink Δ1) and collective dissemination (pink Δ7). Contacts between several structures (pink Δ6). Stellate structures (C): several multicellular chains with a collective cell migration pattern of invasion and protrusive leading front with one (Δ9) or multiple leading cells (Δ15) with invadopodia. Multicellular invasive chains with 1–2 cells in diameter (Δ13) or broad masses of cells (Δ18). Secondary branches at the lateral margin of the invasive chains (Δ12). Cell dissemination (pink Δ7,14,17). Contacts between several structures (pink Δ22,24). (1μM **5-FU** + 0.01μM **4-HC**): Mass structures (A): round morphology (1–3), irregular morphology exhibiting collective cell migration as chains of few cells with smooth borders (Δ5), buds (Δ4), and structures with single-cell protrusions (Δ14,15). Multicellular streaming (Δ7,9). Dissemination of single tumor cells (pink Δ11) and group of cells (pink Δ12). Pseudo-Stellate Mass structures (B): multicellular collective protrusive migration with leading cells with invadopodia (Δ3) or leading buds (Δ9). Dissemination of group of cells (pink Δ17). Stellate

structures (C): multicellular chains with a collective cell migration pattern of invasion and protrusive leading front with invadopodia (Δ5,7), or leading buds (Δ10). Multicellular invasive chains with few cells in diameter (Δ13) or broad masses of cells (Δ13,20). Single and collective cell dissemination (pink Δ27 of the latter). Multicellular chains with an uncoordinated arrangement of the component cells (pink Δ11), contacts (pink Δ15,26) mainly fusions (pink Δ24,25,27). The experiment was repeated at least three times with similar 231/LM2-4 invasiveness inhibitory effect by the treatments. The best multicellular structures were obtained with Matrigel™ matrix basement membrane (BD BioSciences, Cat. 354234-Lot 88482) with a high concentration (10mg/ml).

potential by us and others [59,60], while an article by Mai *et al.* [61] suggests that a "granularity" phenotype, such as the one we observed in treated 3D structures is associated with loss of cell-cell contact but not with invasiveness.

## Discussion

Our results provide a possible explanation for a previously reported paradoxical observation, namely, why a particular metronomic chemotherapy regimen, in this case that was used to treat a triple negative breast cancer xenograft–had only a very modest effect on delaying orthotopic primary tumor growth, but nevertheless caused major inhibition of established metastatic disease in a postsurgical model. The explanation is that the metronomic chemotherapy treatment may mediate anti-invasive effects at the tumor periphery/invasive borders of tumors, and this could include metastases, in the absence of acute massive or even modest cytotoxic effects on the tumor mass. As such, this could be the basis for the impact of metronomic chemotherapy in suppressing the extent of micrometastases detected in mice with primary MDA-MB-231/LM2-4 tumors treated with metronomic UFT alone, metronomic CTX alone, and especially when the two drugs are combined and administered concurrently [13]. While this effect can provide a basis for explaining how the therapy can have an adjuvant therapy effect on early stage micrometastatic disease, it is not as clear whether the apparent anti-invasive effects of the UFT+CTX metronomic chemotherapy protocol and either UFT alone or CTX alone contributed to the observed striking impact of greatly prolonging the survival of mice when treatment was only initiated several weeks after primary tumor resection, i.e., at a time when metastases were already established. It is possible that the anti-invasive effects at the tumor periphery may serve to limit further expansion of tumor growth, and metastatic disease progression, while at the time the prolonged daily therapy could cause a gradual and cumulative tumor cell loss by mechanisms such as direct tumor cell kill via necrosis, as discussed below [62,45], inhibition of angiogenesis, or activating innate immune cell effector activity [7–11].

The current *in vivo* experiment was conducted under the exact same conditions as the aforementioned study. These studies differ only with respect to endpoint, which in our

**Table 1. Quantitative analysis of the morphological phenotypes displayed by 231/LM2-4 multicellular structures grown in a 3D lrECM "on-top" assay using Matrigel as barrier.**

| Group | Mass | 3D Morphology | | Structuresnumber | x10 fields analyzed |
| | | Pseudo-Stellate Mass | Stellate | | |
|---|---|---|---|---|---|
| Control | 49 (33.8%) | 10 (6.9%) | 86 (59.3%) | 145 | 56 |
| 5-FU | 159 (61%) | 46 (17.6%) | 56 (21.4%) | 261 | 88 |
| 4-HC | 131 (59.5%) | 44 (20%) | 45 (20.4%) | 220 | 95 |
| 5-FU + 4-HC | 136 (53.3%) | 46 (18%) | 73 (28.6%) | 255 | 101 |

The 231/LM2-4 cells were culture without drugs (Control) or with 1μM 5-FU, 0.01μM 4-HC or the combination 1μM 5-FU + 0.01μM 4-HC for 6 days. The medium was changed every single day for six days. Chi-square analysis show statistical difference for control *versus* 5-FU, 4-HC and 5-FU+4-HC (p<0.05). No significant difference was found with comparisons involving 5-FU, 4-HC and 5-FU + 4-HC (Chi-square analysis; p>0.05).

previous experiment was the maximum tumor volume ethically permitted, 1,700mm$^3$, while in the current experiment was around 1,300mm$^3$ in all groups, reasoning that the control tumors at this size could reflect an earlier step in the metastatic cascade and that standardization by size could minimize the influence of microenvironmental factors such as hypoxia or insufficient blood flow that are size dependent.

To implicate the anti-invasive effects of the metronomic UFT plus CTX doublet combination, or of the respective monotherapies, we undertook six different assays, four *in vivo* and two *in vitro*: evaluating necrosis in the tumor core vs. the tumor periphery, evaluating vascular density, again in the tumor core vs. tumor periphery, assessing p-Met[Y1003], assessing impact on the extent of tumor collagen deposition in activity; the two *in vitro* assays used as surrogates for *in vivo* tumor cell invasion were a transwell migration assay and a 3D matrigel assay.

In the evaluation of necrosis in the tumor core vs. the tumor periphery, our results suggest that in the tumor periphery, both UFT and CTX, when used as single agents, caused necrosis, and a significant increase in necrosis was observed in the UFT+CTX treated tumors with respect to control group. The level of necrosis in the invasive border sections of the tumors might be indicative of a meaningful contribution mediated by each experimental condition to cell death through necrosis. These results additionally indicate that the observed lack of effect of the UFT treatment in tumor volume, as measured externally using a calliper, may be misleading. Therefore, the volume of a viable tumor should be more reliably calculated from the total tumor volume minus the volume of the necrotic tumor [63]. In addition, local invasion changes may be masked when only assessing tumor volumes.

In parallel, we evaluated the effect of the different treatments on tumor vasculature and we determined an anti-angiogenic effect of low-dose metronomic CTX and the UFT+CTX combination. This reduction in vascular tumor density is associated with a reduction in tumor volume in both groups of treatment.

The expression and cellular distribution of c-Met and its activated form p-Met[Y1003] was analyzed in untreated and treated 231/LM2-4 tumor xenografts. A specific correlation between c-Met expression and basal breast cancer subtypes was previously established using tumor tissues [64]. Charaffe-Jauffret *et al.* [50] in a gene expression profiling of breast cancer cell lines, belonging to luminal and basal cell subtypes, showed that the discriminator genes between both subtypes are associated with cell adhesion, cell motility and invasion, which might be related to the poorer basal tumor prognosis. These discriminator genes included met and Met protein signalling partners. c-Met protein is involved in cancer progression and metastasis. A significant correlation between c-Met overexpression, aberrant c-Met activation and poor outcome has been found in different solid tumors, including breast cancer [65]. c-Met phosphorylation is an indicator of signalling activation and thus reflects elevated tumor activity compared to unphosphorylated receptor expression [66]. Selection of Y1003, was based on its better immunodetection [67,68] and because high levels of p-Met[Y1003] were associated with malignancy and therapy resistance. For instance, p-Met[Y1003] overexpression in non-small-cell lung cancer patients, correlated with gefitinib resistance and shorter time to progression [69]. In addition, p-Met[Y1003] was detected in human melanomas but not in nevi or normal epidermis [67]. Phospho-Y1003 is responsible for the recruitment of Cbl ubiquitin ligase to c-Met, which facilitates c-Met endocytic trafficking into the lysosomal compartment. c-Met endocytic trafficking is not only necessary for signal attenuation, but it is required for optimal signaling [70,71]. It is important to note that the high expression of c-Met and its activated status in the xenograft model employed here permitted its immune-detection and analysis; however, it is not clear which mechanism accounts for c-Met activation in the tumors. In treated 231/LM2-4 tumor xenografts, also strong c-Met levels, localizing to the nucleus and cytoplasm,

were observed. However p-Met[Y1003] showed only focally immunopositivity in a predominantly nuclear pattern. A significant reduction in p-Met[Y1003] levels were found in the UFT +CTX group compared to control.

In relation with the two *in vitro* assays used as surrogates for *in vivo* tumor cell invasion the transwell migration assay, using collagen type IV as coating, showed a significant inhibition of invasion potential of 231/LM2-4 cells in the presence of the drugs *versus* control conditions. and a 3D matrigel assay. However, the obvious reduction of invasion caused by all treatments *in vitro* was not paralleled by our qualitative assessment of local invasion by vimentin staining in treated *versus* control xenografts. This might reflect the complex and dynamic process of metastasis *in vivo*, in which different players in space and time are involved. By other hand, the 3D matrigel assay showing a detailed morphologic analysis of the 3D structures formed by the 231/LM2-4 cells on Matrigel suggest that metronomic UFT, CTX and the double UFT+CTX treatment may inhibit local invasion of 231/LM2-4 cells, possibly as a result of alterations in cell-cell and cell-ECM interactions.

The results of all aforementioned assays, when considered together, create a reasonable argument for the anti-invasive properties of metronomic chemotherapy, at least for the regimen tested. A case in point: the elevated levels of necrosis (and vascular density reductions) at the tumor periphery caused by metronomic UFT or metronomic CTX and especially the two together on MDA-MB-231/LM2-4 primary orthotopic tumors, and a similar pattern for p-Met [Y1003] inhibition. The necrosis findings at the tumor periphery may be pertinent to the above mentioned question regarding how the metronomic UFT+CTX treatment protocol administered continuously over many months could conceivably gradually diminish the growth and size of established metastases and thus greatly prolong survival.

In summary, our results provide a possible new perspective on how metronomic chemotherapy may mediate inhibition of tumor growth and metastatic disease progression. They also highlight the limitations of relying only on tumor growth/size changes as an indication of anti-tumor activity in preclinical models involving treatment of established primary tumors– lack of activity, or detection of modest activity, does not necessarily preclude more potent activity on micro or macroscopic metastases.

While our *in vivo* results were initially based on a single experiment with a small number of animals per group as shown in Fig 3, we have undertaken additional efforts to support our findings. We undertook these efforts to examine whether the original effect on metastatic disease noted [13] could be extended into a more clinically relevant model of neoadjuvant therapy. In that regard, the evidence for an anti-metastatic potential associated with this approach is now supported by subsequent examination of its effect in this setting (S1 Appendix). Using this approach a significant effect was noted with respect to bone metastasis. While the levels of lung metastases were lower in the treatment groups CTX and UFT+CTX, these did not reach statistical significance.

In summary, the *in vivo* studies indicate that the doublet treatment with metronomic UFT +CTX is associated with a significant increase in tumor necrosis at the invasive border areas and a significant reduction in vascular density that are in turn associated with tumor growth inhibition. In addition, all treatments reduced the amount of intratumoral collagen deposition, being more noticeable in the UFT+CTX treated tumors. Finally, UFT+CTX caused a significant reduction in p-Met[Y1003] levels compared to control. The impact of these effects on the invasive capacity of tumor cells cannot be directly determined from our data, but all of these features, especially taken together, could conceivably contribute to the anti-metastatic effects of the doublet UFT+CTX metronomic therapy treatment reported in our previous study [13].

## Materials and methods

### Drugs and schedule

Tegafur, uracil and the vehicle HPMC were supplied by Taiho Pharmaceutical Co., Ltd. CTX (Baxter Oncology GmbH) was purchased from the institutional Pharmacy, and was reconstituted as per manufacturer´s instructions to a stock concentration of 20mg/ml. UFT, a 5-FU oral prodrug, was prepared fresh daily, immediately prior to use, by adding an aqueous solution of uracil containing 0.1% of HPMC to tegafur in a molar ratio of 4:1, and was administered at 15mg/Kg/d by gavage during the length of the experiment. The CTX treatment was initiated with a bolus intraperitoneal injection of 100mg/Kg body weight (approximately one third of the maximum tolerated dose) on the first day of treatment, and then switched to oral administration, via drinking water, to provide an estimated dose of 20mg/Kg/d. This treatment schedule was in line, with the "fast" and "slow" metronomic regimen, consisting of daily oral low dose therapy, interspersed every 6 weeks, with a bolus intraperitoneal injection of the drug [13]. The vehicle 0.1% HPMC, as a control, was administered every day by gavage.

A protracted continuous low-dose exposure to the chemotherapeutic drugs, 5-FU, 4-HC and the 5-FU+4-HC combination treatment, was used in the *in vitro* studies. The drug concentrations used in each assay are indicated in the corresponding figure legends. 5-FU (Sigma-Aldrich) was diluted in culture medium immediately before use. The toxic metabolite/alkylating agent 4-hydroperoxycyclophosphamide (4-HC) undergoes spontaneous reduction in solution to 4-hydroxycyclophosphamide, which is the first active metabolite formed in the metabolism of CTX. 4-HC was a gift of Dr S.M. Ludeman (Duke University, Durham, NC). It was stored as a powder at -20˚C and prior to use was dissolved at 4˚C in distilled water and filtered in sterile conditions.

### Cells and culture conditions

The highly metastatic MDA-MB-231 human breast cancer variant 231/LM2-4 has been used [13]. Cell line authentication was carried out by genotyping using Illumina mouse linkage panel and confirmed to be human in origin. Routine mycoplasma screening is carried out in-house using commercial kits, which confirmed the cell line is mycoplasma free. The cells were grown in RPMI 1640 supplemented with 5% heat-inactivated FBS, at 37˚C in a humidified incubator and 5% $CO_2$.

### *In vivo* tumor growth assessment

$2x10^6$ 231/LM2-4 cells in a 50µl volume were orthotopically implanted into the right inguinal mammary fat pad of CB17 severe combined immunodeficient (SCID) mice [13]. Surgical procedures were undertaken using inhaled isoflurane, supplemented with 100% $O_2$, as anesthesia technique and Buprenorphine injectable (temgesic®) was used as analgesic. Following surgery, the mice were placed under an infrared lamp to prevent hypothermia and were injected subcutaneously with Buprenorphine injectable for pain control. Weekly caliper measurement were performed to determine tumor growth and tumor volume ($mm^3$) was assessed using the formula $a^2$ x b/2, where a and b represent the smallest and largest tumor diameter, respectively. All mice were randomized just before initiation of treatment. Treatment was initiated when the primary intra-mammary fat pad tumor attained a size of 160-190$mm^3$. The mice were sacrificed when the tumors reached an average size of approximately 1,300$mm^3$. Animal euthanasia was conducted by means of cervical dislocation, immediately after the animals were anesthetized by isoflurane inhalation.

In order to ensure the greatest animal welfare throughout the experimental process, all *in vivo* experiments were carried out with the approval of Sunnybrook Research Institute Animal Care Committee and followed the principles and guidelines of the Canadian Council on Animal Care.

### Histochemistry and Immunohistochemistry

The resected mouse tumors were dissected into two parts: the central region and the invasive border of the tumors. Both pieces of each tumor were immediately fixed in 10% buffered formalin for 24 hours, and stored in 70% ethanol prior to processing and embedding in paraffin. Sections of 5μm were deparaffinized and several histochemical staining were carried out. The necrotic areas of the tumors were assessed by H&E stain. The collagen content of the tumors was analyzed by a modified Masson´s trichrome staining in accordance with the protocol of MacLean *et al.* [72]. For immunohistochemistry, procedures were as follow: mouse mAb anti vimentin (clone V9), Dako Cytomation (≠M0725) [microwave 10min., citrate buffer, pH 6; 1:50 overnight 4˚C]. Rabbit anti Ki67, Vector laboratories VP-K451 [microwave 10min., citrate buffer, pH 6; 1:1000 overnight 4˚C. Double immunostaining with goat polyclonal anti CD31/PECAM-1 (M-20), Santa Cruz (#SC-1506) and Rabbit mAb anti VEGFR2 (55B11), Cell Signaling (#2479) [water-bath 96˚C 20min., Tris-EDTA, pH 9; 1:100 overnight 4˚C and 1:100, 1hour room temperature, respectively]. Mouse mAb anti c-Met (clone 3D4), Invitrogen (≠18–7366) [microwave 15min., citrate buffer, pH 6; 1:50 overnight 4˚C]. Rabbit anti c-Met [pY1003], Invitrogen (≠44-882G) [microwave 15min., citrate buffer, pH 6; 1:500 overnight 4˚C]. In the immunoenzymatic staining, DAB was used as peroxidase substrate (Histostain® Plus Broad Spectrum (DAB), 85–9643). In the immunofluorescence staining donkey anti-goat Fluor 488 (1:200; Santa Cruz Biotechnology) for 30min. and goat anti-rabbit Cy3 (1:200; Jackson ImmunoResearch) for 30min. were used. The effect of different treatments on vascular density was determined by double CD31/VEGFR2 immunofluorescence labeling [44]. Positive blood vessels showed staining for the pan-endothelial cell marker CD31 and for VEGFR2. Seven fields per tumor, from the central region and the invasive border, were counted in areas of highest vascular density ("hot spots") using a Leica DMLB microscope (Q imaging QICAM fast 1394 digital camera) at x20 magnification. The average vessel count/field was determined. Immunohistochemistry was also performed to study the expression of the invasion marker c-Met and p-Met[Y1003]; the immunohistochemistry analysis were evaluated semiquantitatively by assigning an H-score (or "histo" score) to tumor samples of 0, 1+, 2+, or 3+ [73,74]. The H-score is based on a predominant staining intensity and the relative area of cells of any given intensity of staining in a proportion to the remaining sample, so the percentage of cells at each staining intensity level is calculated for each sample, and finally, an H-score is assigned using the following formula: [1 × (% cells 1+) + 2 × (% cells 2+) + 3 × (% cells 3+)]. The final score, ranged from 0 to 300. Based on the H-scores we performed statistical analysis using Student's t-test assuming unequal variance. The staining results were interpreted by two expert independent pathologists (DH and GW) blinded to the different treatment groups.

### *In vitro* viability assay

An *in vitro* viability assay was performed on a single-cell suspension of 231/LM2-4 cells plated on 96-well plates and allowed to attach overnight. Cells ($1x10^3$ cells/well in 200μl of medium) were untreated and treated for 6 days with several concentrations of 5-FU (0.1, 0.5, 1 and 10μM), 4-HC (0.01, 0.05, 0.1 and 1μM) and the 1μM 5-FU+0.01μM 4-HC combination. The cells were incubated at 37˚C in 5% $CO_2$, and every 24 hours the medium was replaced with a fresh solution to maintain a constant concentration of the drugs during the whole length of

the experiment, as designed by Bocci *et al*. [75]. At the end of the experiment, cells were incubated, with MTS [3-(4,5-dimethylthiazol-2-yl)-5-(3-carboxymethoxyphenyl)-2-(4-sulfophenyl)-2H tetrazolium] tetrazolium salt (Promega PR-G1112) and the electron coupling reagent phenazinemethosulfate, following manufacturer instructions. MTS is converted by dehydrogenase enzymes of metabolically active cells into formazan. The absorbance of the formazan at 490nm is directly proportional to the number of living cells. Correcting absorbance at 490nm with the reference wavelength at 650nm eliminates any kind of background.

## Drug response in a transwell chemoinvasion assay using type IV collagen

The type IV collagen acquired from BD Biosciences (Catalog number 354233) was diluted to 0.2mg/ml in RPMI without FBS. The upper surfaces of the transwell cell culture inserts containing membrane (8.0μm pore size) were placed on a companion Multiwell$^{TM}$ 24 well plate (Becton Dickinson, Cat.3504), and were coated by adding 50μl of diluted collagen and left overnight at 37˚C. The 231/LM2-4 cells were plated on 3cm tissue-culture plates and allowed to attach overnight. Next day, the medium was changed and cells were untreated or treated with 1μM 5-FU, 0.01μM 4-HC and the 1μM 5-FU+0.01μM 4-HC combination, the medium being replaced every 24 hours with fresh solution. After four days, the cells were starved in RPMI 1640+0.1% BSA+0.2% FBS medium, with or without drugs, and incubated for one day longer at 37˚C, 5% CO$_2$. On the sixth day the subconfluent cells were detached with trypsin, resuspended in RPMI 1640+5% FBS and washed twice, once with RPMI 1640 without FBS and once with RPMI 1640+0.1% BSA+0.2% FBS medium, with or without drugs. Finally, the cells were resuspended in RPMI 1640+0.1% BSA+0.2% FBS medium, with or without drugs, and a concentration of 5x10$^5$ cell/ml and 0.2ml of the cellular suspensions was placed onto the upper surface of the inserts; as a chemoattractant, 0.6ml of RPMI 1640+0.1% BSA+5% FBS medium was added to the well containing the insert. The plates were left at 37˚C, 5% CO$_2$, and 24–30 hours was the time allowed for invasion to occur. The culture medium was then removed and the inserts containing invading tumor cells were properly washed with PBS on the upper and lower surfaces, fixed with 100% methanol for 20min. and stained for 20min. with a solution of 0.1% crystal violet prepared in 20% methanol, and distained with distilled water. A cotton swab was used to remove cells and collagen present on the upper surface of the inserts. The dry and clean membranes were cut, placed on a microscope slide, with the surface containing the invading cells face up, and mounted. The number of tumor cells was quantified by counting eight random fields per sample on a grid, under light microscopy (bright field) at x20 magnification.

## Drug response in the three-dimensional laminin-rich extracellular matrix (Matrigel$^{TM}$) "on-top" embedded culture assay

Pre-cooled 3cm tissue culture plates were covered with 500μl of 100% Matrigel$^{TM}$ matrix Basement Membrane (BD Biosciences, Cat.354234) and then placed at 37˚C to allow the Matrigel to solidify [76]. Subconfluent monolayers of low passage 231/LM2-4 cells, kept under standard culture conditions, were trypsinized with a small amount of Trypsin-EDTA solution 0,05%, resuspended in RPMI+5% FBS, passed three times through a 10ml syringe 18G needle, washed to remove trypsin, and finally counted. Cells were centrifuged and resuspended in the Mammary Epithelium Growth Media (MEGM) containing 10ng/ml human epidermal growth factor (hEGF), 5μg/ml insulin, 0.5μg/ml hydrocortisone, and supplemented with Bovine Pituitary Extract (BPE), to which 2% Matrigel was added. After cell resuspension, 400μl (30,000 cells) were plated as a single cell suspension on the plates covered with 100% Matrigel$^{TM}$ matrix, to which 400μl of MEGM+BPE media were added 30min. before plating. Following careful

swirling, the cells were allowed to settle for 5min. and then swirled again, to ensure a homogenous distribution on the plates. Subsequently, 200μl of cold MEGM supplemented mammary medium containing 10% of Matrigel was added to form an overlay [56].

After 24 hours of incubation at 37°C the medium was removed and 1ml of fresh MEGM supplemented medium, containing 2% of Matrigel with or without the different drugs, was added. The cells were culture for 6 days and the medium with/without drugs was changed every day. The best multicellular structures were obtained using Matrigel™ matrix basement membrane (BD BioSciences, Cat. 354234-Lot 88482) with a high matrigel concentration of 10 mg/ml.

## Supporting information

**S1 Fig. Evaluation of body weight in mice untreated and treated with metronomic UFT, CTX or a combination of both drugs.**
(TIF)

**S2 Fig. Analysis of vascular density.** Dual immunofluorescence staining for CD31 (green) and VEGFR2 (red) in 231/LM2-4 tumor samples.
(TIF)

**S3 Fig. Vimentin immunostaining in 231/LM2-4 xenografts.** Vimentin staining shows the invasive front of the tumor and the tumor cells invading the stroma. The primary antibody is specific to human vimentin. The negative control of the immunostaining is therefore the mouse tissue itself.
(TIF)

**S4 Fig. Trichrome histochemical staining of intratumoral collagen.** We used a grading system composed of mild/focal (grade 1), moderated (grade 2), and extensive (grade 3) collagen deposition.
(TIF)

**S5 Fig. Side by side comparison of p-Met[Y1003] and Ki67 staining in 231/LM2-4 tumor xenografts.** Areas of positive staining overlap.
(TIF)

**S6 Fig. Expression and cellular distribution of c-Met in treated 231/LM2-4 tumor xenografts.** A strong nuclear and cytoplasmic expression of c-Met was observed in all treatment groups, with no noticeable difference.
(TIF)

**S7 Fig. Assessment of the invasive capacity of 231/LM2-4 cells treated with 5-FU or 4-HC by a 3D lrECM "on-top" assay using Matrigel as barrier.** Representative examples of the different morphological phenotypes of the multicellular structures. (1μM **5-FU**): Mass structures (A): round morphology (1–4), collective cell migration as chains of few cells with smooth borders (Δ11,16), buds (Δ6), or as disorganized masses (Δ22). Single-cell protrusions (Δ26,27). Multicellular streaming with no apparent junction contacts (Δ13,17). Dissemination of single tumor cells (pink Δ20,28) and group of cells (pink Δ19,29). Pseudo-Stellate Mass structures (B): multicellular collective protrusive migration with leading cells with invadopodia (Δ3,10,14) or leading buds (Δ11), and a loose assembly of individual round cells in multicellular structures (Δ1,4,6,7). Dissemination of single tumor cells (pink Δ9) and group of cells (pink Δ17). Contact (pink Δ12) and fusion (pink Δ15) between different structures. Stellate structures (C): protrusive leading front with invadopodia (Δ11) or leading buds (Δ3,4,14).

Multicellular invasive chains with 1–2 cells in diameter (Δ2) or broad masses of cells (Δ18). Collective cell dissemination (pink Δ15). An uncoordinated arrangement of the component cells in some multicellular chains (pink Δ13,17), contacts (pink Δ8,19), fusions (images 16,20,22) between different structures to form a large stellate structure. (0.01μM **4-HC**): Mass structures (A): round morphology (1–3), collective cell migration as chains of few cells with smooth borders (Δ7), buds (Δ5), or as disorganized masses (Δ19). Single-cell protrusions (Δ21,26). Multicellular streaming with no apparent junction contacts (Δ10,11). Dissemination of single tumor cells (pink Δ16,25) and group of cells (pink Δ17). Pseudo-Stellate Mass structures (B): multicellular collective protrusive migration pattern containing leading cells with invadopodia (Δ5,8) or leading buds (Δ1), and a loose assembly of individual round cells in multicellular structures (Δ4). Dissemination of single tumor cells (pink Δ2). Fusion between different structures (pink Δ18). Stellate structures (C): protrusive leading front with invadopodia (Δ17) or leading buds (Δ10). Multicellular invasive chains consisted of one or two cells in diameter (Δ12) or broad masses of cells (Δ8). Single cell dissemination (pink Δ16). An uncoordinated arrangement of the component cells in some multicellular chains (pink Δ21). Contacts (pink Δ14,23) or more commonly fusions (images 13,19,22) between different structures to form a large stellate structure.
(TIF)

**S1 Table. Assessment of peritumoral and intratumoral collagen deposition in paraffin tumor sections.**
(DOCX)

**S2 Table. Assessment of p-Met[Y1003] in paraffin tumor sections.**
(DOCX)

**S1 Appendix. Assessment of the anti-metastatic effect associated with UFT+CTX therapy in the neoadjuvant setting in 231/LM2-4 breast cancer model.**
(DOCX)

## Acknowledgments

We are grateful to Teiji Takechi, from Taiho Pharmaceutical, for providing UFT. We wish to thank Helen Coates for doing the Trichrome staining, Mariana Capurro for her inestimable scientific advices, Cristopher Jedeszko for technical help and advice, Celia García-Hernández for her generous artistic contribution, William Sears (Department of Population Medicine, University of Guelph) for statistical guidance and analysis and to Cassandra Cheng for her generous secretarial assistance.

*In memoriam of Helen Coates.*

## Author Contributions

**Conceptualization:** Raquel Muñoz, Robert S. Kerbel.

**Data curation:** Raquel Muñoz, William Cruz-Muñoz, Ping Xu.

**Formal analysis:** Raquel Muñoz, Denise Hileeto, William Cruz-Muñoz, Ping Xu.

**Funding acquisition:** Alicia Viloria-Petit, Robert S. Kerbel.

**Investigation:** Raquel Muñoz, Denise Hileeto, William Cruz-Muñoz, Geoffrey A. Wood, Ping Xu, Shan Man.

**Methodology:** Raquel Muñoz, Shan Man, Alicia Viloria-Petit.

**Project administration:** Raquel Muñoz, Shan Man, Alicia Viloria-Petit, Robert S. Kerbel.

**Resources:** Ping Xu, Shan Man.

**Supervision:** Raquel Muñoz, Alicia Viloria-Petit, Robert S. Kerbel.

**Validation:** Raquel Muñoz, William Cruz-Muñoz, Alicia Viloria-Petit.

**Visualization:** William Cruz-Muñoz, Shan Man.

**Writing – original draft:** Raquel Muñoz, Denise Hileeto, Alicia Viloria-Petit.

**Writing – review & editing:** Raquel Muñoz, William Cruz-Muñoz, Alicia Viloria-Petit, Robert S. Kerbel.

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
