## [Decision Letter · Decision Letter 0]

2 Jul 2019

PONE-D-19-16716

Impact of metronomic UFT, cyclophosphamide, or when combined on mediators of breast cancer dissemination in vivo and 3D invasiveness in vitro

PLOS ONE

Dear Dr. Kerbel,

Thank you for submitting your manuscript to PLOS ONE. After careful consideration, we feel that it has merit but does not fully meet PLOS ONE’s publication criteria as it currently stands. Therefore, we invite you to submit a revised version of the manuscript that addresses the points raised during the review process by the two Reviewers, both experts in the metronomic field.

We would appreciate receiving your revised manuscript by Aug 16 2019 11:59PM. To enhance the reproducibility of your results, we recommend that if applicable you deposit your laboratory protocols in protocols.io, where a protocol can be assigned its own identifier (DOI) such that it can be cited independently in the future. For instructions see: http://journals.plos.org/plosone/s/submission-guidelines#loc-laboratory-protocols

We look forward to receiving your revised manuscript.

Kind regards,

Francesco Bertolini, MD, PhD

Academic Editor

PLOS ONE

Journal Requirements:

2. To comply with PLOS ONE submissions requirements, in your Methods section, please provide additional information regarding the experiments involving animals and ensure you have included details on (1) methods of sacrifice at the end of the study, (2) methods of anesthesia and/or analgesia, and (3) efforts to alleviate suffering.

Reviewers' comments:

Reviewer's Responses to Questions

**Comments to the Author**

1. Is the manuscript technically sound, and do the data support the conclusions?

Reviewer #1: Yes

Reviewer #2: Partly

2. Has the statistical analysis been performed appropriately and rigorously? 

Reviewer #1: No

Reviewer #2: No

3. Have the authors made all data underlying the findings in their manuscript fully available?

Reviewer #1: Yes

Reviewer #2: Yes

4. Is the manuscript presented in an intelligible fashion and written in standard English?

Reviewer #1: Yes

Reviewer #2: No

5. Review Comments to the Author

Reviewer #1: In this paper, Munoz et al provide a possible explanation for a previous work by the same group reporting paradoxical effect of metronomic chemotherapy on primary tumor growth and metastatic disease in a TNBC model. The authors evaluate the effect of low-dose CTX and UFT, and their combination, on vascular density, collagen deposition and c-Met in the primary tumor setting via histochemistry/immunohistochemistry. Moreover, the authors evaluated the in vitro effect of continuous low-dose of active drug metabolites using a transwell migration assay and a 3D matrigel assay.

In order to better appreciate the manuscript, some minor revisions are needed:

1. The title is unclear, please rephrase it.

2. The introduction, even if it is clear and well written, is too long. It must be shortened.

3. In the results sections, Munoz and colleagues also include a discussion about the data. The authors should comment the data specifically in the discussion part. Furthermore, the titles of the paragraphs in the results section must be explicative of the data presented. Do not use general terms as “effects” and “assessment”, please focus more on the take-on message.

4. In general, I suggest to the authors to present the data reported in the tables with graphs. The raw data could be uploaded as supplementary materials. Also please, provide statistical analysis and a legend for the data in table 1.

5. The main scope of this manuscript is to shed light on the mechanism beyond the observation that UFT, either alone or in combination with CTX, had a modest effect on primary tumor growth, but is able to inhibit metastatic spread. My main concern regards the fact that the authors based most of the evidence from a previous paper (Munoz R et al, 2006). Have the authors performed new experiments to evaluate lung metastasis at mice sacrifice or after tumor resection? If not, they should provide them to corroborate the previous evidences.

Reviewer #2: Munoz et at reexamine a metronomic chemo drug schedule in a triple negative orthotopedic breast center model that they first described in 2006 (Ref #13), to determine how/why uracil-tegafur-cyclophosphamide (UFT + CTX) inhibits micro-metastasis development but has little inhibitory effects on primary tumor growth. They now find that, although the combination drug treatment has little effect on primary tumor volume (Fig. 1), it significantly increases necrosis and decreases vascularity in invasive border sections of the tumor (Fig. 2). Further, it decreases phosphorylation of the growth factor receptor c-Met, which contributes to tumor growth and metastasis (Fig. 5). Most interesting is the finding, using cell culture models and a trans-well invasion assay, that drug treatment inhibits tumor cell migratory and inhibitory potential but not tumor cell viability. Together, these finding provide a substantial advance in explaining the apparent discrepancy raised by findings in the 2006 study.

Specific concerns include the following:

1. Why did the authors not confirm their 2006 finding of inhibition of micrometastases by the UFT + CTX combination? Tumor cell models can drift/evolve over time (13 years, in this case), and there is some concern that the in vivo findings with primary tumors, reported here, might not be directly applicable to the earlier findings of inhibition of metastasis in what may only nominally be the same tumor model. Further, if the prior metastasis study was performed in a post-surgical model, then the current findings about impact of UFT + CTX on metastatic potential in a non-surgical model may not be directly relevant.

2. In several places, conclusions stated in the text need to be based on a more rigorous implementation of statistical analysis. For example:

- Line 177 - Fig. 1 conclusion of delayed tumor growth needs statistical support. (Also, mark day of first drug treatment along the x-axis)

- Line 281 – Any conclusion of reduced invasion into adjacent soft tissue requires statistical analysis of Table 1 results.

- Line 300 – Similarly, the conclusion stated in the text of reduced collagen deposition in all treatment groups requires statistical analysis of Table 2 data. Text on line 302 indicates that is not significant.

- - Line 387 – Staining scores should be listed for each individual tumor in a supplemental Table, similar to the format of Table 2.

3. The entire work is apparently based on a single in vivo experiment (Fig. 1) with only 4-5 tumors per group. This limitation should be explicitly noted, e.g., in Discussion.

4. What is the impact of drug treatment on MMP expression – in vitro or in vivo (line 464)?

5. Is the in vitro HC concentration really 10nM? Are there any prior studies showing a biological effect of HC at such a low concentration? That concentration is 100-500-fold lower that typically reported anti-tumor activities for HC.

6. Minor Points

- Shorten Introduction by editing.

- Line 61 – Correct text to indicate clinical trial metronomic (?) studies

- Line 381 – Arrow not visible on Figure

- Table 3 – Specify drug concentration. Same for Figure S6, etc. (see line 631).

- Line 617 ff. Specify sources for HC.

6. PLOS authors have the option to publish the peer review history of their article (what does this mean?). If published, this will include your full peer review and any attached files.

Reviewer #1: No

Reviewer #2: No

---

## [Author Response · Author response to Decision Letter 0]

22 Aug 2019

Please see uploaded Response to Reviewers letter at the end of the PDF

---

## [Decision Letter · Decision Letter 1]

4 Sep 2019

[EXSCINDED]

Suppressive impact of metronomic chemotherapy using UFT and/or cyclophosphamide on mediators of breast cancer dissemination and invasion

PONE-D-19-16716R1

Dear Dr. Kerbel,

We are pleased to inform you that your manuscript has been judged scientifically suitable for publication and will be formally accepted for publication once it complies with all outstanding technical requirements.

With kind regards,

Francesco Bertolini, MD, PhD

Academic Editor

PLOS ONE

Additional Editor Comments (optional):

Reviewers' comments:

Reviewer's Responses to Questions

**Comments to the Author**

1. If the authors have adequately addressed your comments raised in a previous round of review and you feel that this manuscript is now acceptable for publication, you may indicate that here to bypass the “Comments to the Author” section, enter your conflict of interest statement in the “Confidential to Editor” section, and submit your "Accept" recommendation.

Reviewer #1: All comments have been addressed

Reviewer #2: All comments have been addressed

2. Is the manuscript technically sound, and do the data support the conclusions?

Reviewer #1: Yes

Reviewer #2: Yes

3. Has the statistical analysis been performed appropriately and rigorously? 

Reviewer #1: Yes

Reviewer #2: Yes

4. Have the authors made all data underlying the findings in their manuscript fully available?

Reviewer #1: Yes

Reviewer #2: Yes

5. Is the manuscript presented in an intelligible fashion and written in standard English?

Reviewer #1: Yes

Reviewer #2: Yes

6. Review Comments to the Author

Reviewer #1: The authors through their response and edits have addresses all comments and concerns. I have no further suggestions or corrections.

Reviewer #2: (No Response)

7. PLOS authors have the option to publish the peer review history of their article (what does this mean?). If published, this will include your full peer review and any attached files.

Reviewer #1: No

Reviewer #2: No

---

## [Editor Report · Acceptance letter]

11 Sep 2019

PONE-D-19-16716R1 

Suppressive impact of metronomic chemotherapy using UFT and/or cyclophosphamide on mediators of breast cancer dissemination and invasion 

Dear Dr. Kerbel:

I am pleased to inform you that your manuscript has been deemed suitable for publication in PLOS ONE. Congratulations! Your manuscript is now with our production department. 

With kind regards,

on behalf of

Dr. Francesco Bertolini 

Academic Editor

PLOS ONE